# Analysis of Leukocyte Subpopulations by Flow Cytometry during Hospitalization Depending on the Severity of COVID-19 Course

**DOI:** 10.3390/biomedicines11102728

**Published:** 2023-10-08

**Authors:** Elżbieta Rutkowska, Iwona Kwiecień, Ewa Pietruszka-Wałęka, Ewa Więsik-Szewczyk, Piotr Rzepecki, Karina Jahnz-Różyk

**Affiliations:** 1Laboratory of Flow Cytometry, Department of Internal Medicine and Hematology, Military Institute of Medicine Warsaw—National Research Institute, 04-141 Warsaw, Poland; ikwiecien@wim.mil.pl; 2Department of Internal Medicine, Pulmonology, Allergology and Clinical Immunology, Military Institute of Medicine Warsaw—National Research Institute, 04-141 Warsaw, Poland; epietruszka@wim.mil.pl (E.P.-W.); ewiesik-szewczyk@wim.mil.pl (E.W.-S.); kjrozyk@wim.mil.pl (K.J.-R.); 3Department of Internal Medicine and Hematology, Military Institute of Medicine Warsaw—National Research Institute, 04-141 Warsaw, Poland; przepecki@wim.mil.pl

**Keywords:** SARS-CoV-2 infection, flow cytometry, lymphocytes, leukocytes, T lymphocytes, B lymphocytes, neutrophils

## Abstract

The mechanisms underlying the immune response to coronavirus disease 2019 (COVID-19) and the recovery process have not been fully elucidated. The aim of the study was to analyze leukocyte subpopulations in patients at significant time points (at diagnosis, and 3 and 6 months after infection) selected according to the analysis of changes in the lungs by the CT classification system, considering the severity of the disease. The study groups consisted of severe and non-severe COVID-19 patients. There was a significant decrease in CD8+ T cells, NK and eosinophils, with an increasing percentage of neutrophils during hospitalization. We noticed lower levels of CD4 and CD8 T lymphocytes, eosinophils, basophils, and CD16+ monocytes and elevated neutrophil levels in severe patients relative to non-severe patients. Three months after infection, we observed higher levels of basophils, and after 6 months, higher CD4/CD8 ratios and T cell levels in the severe compared to non-severe group. Non-severe patients showed significant changes in the leukocyte populations studied at time of hospitalization and both within 3 months and 6 months of onset. The CT CSS classification with parameters of the flow cytometry method used for COVID-19 patients determined changes that proved useful in the initial evaluation of patients.

## 1. Introduction

The 2019 coronavirus disease (COVID-19) pandemic caused by the severe acute respiratory syndrome 2 coronavirus (SARS-CoV-2) has led to serious health and economic consequences worldwide [1,2].

Most people infected with SARS-CoV-2 showed slight symptoms, while some demonstrated severe symptoms that required hospitalization, often due to deterioration of lung function [3]. The most common complications include acute respiratory distress syndrome, neurological and circulatory ailments, general weakness of the body and even multiple organ failure [4]. During the inflammatory response, an increase in pro-inflammatory cytokines and activation and exhaustion of immune cells was observed [5]. 

A comprehensive understanding of COVID-19, associated risks and protective factors is essential to prevent infection, progression and adverse disease outcomes. Based on current research results, risk factors for developing COVID-19 in adults include advanced age, overweight, male gender, and the presence of underlying diseases such as chronic obstructive pulmonary disease, cardiovascular diseases and hypertension [6,7]. With the onset of the pandemic, there has been a recognized need to identify risk factors for severity and mortality associated with COVID-19, finding potential predictors of disease severity and outcome in individual patients [8]. Patients with risk factors can be identified earlier, and intensive surveillance or preventive treatment may be initiated to improve prognosis, and more aggressive or more specific targeted therapy can be administered [9].

Numerous studies that analyzed the immune response against SARS-CoV-2 in convalescents indicate the possibility of obtaining immune protection after natural infection [10,11,12]. However, little is known about the comprehensive impact of infection on the immune system after recovery. Lymphocytes and their subpopulations such as T cells, B cells and NK cells play an important role in the proper and long-term functioning of the immune system [13,14]. SARS-CoV-2 infection changes the total number and distribution of lymphocytes and their subsets according to disease severity, indicating a potential relationship between the change in the lymphocyte subset and the pathogenic mechanisms of the virus [15]. Studies show a marked decrease in peripheral lymphocytes as well as changes in subsets of T cells in COVID-19 patients [16]. Profiling T and B cell responses appears to be a good prognostic biomarker for disease development in COVID-19 patients [17]. Alterations in T cell numbers and phenotype develop early and correlate with the recovery process, but are also relatively impaired in severe disease through intense activation and lymphopenia [18,19].

We point out that flow cytometry is an important research and diagnostic method for the rapid acquisition of biological data without the need for expensive, highly advanced equipment. It is also an excellent tool for assessing cellular heterogeneity and lymphoid and myeloid activation states based on surface antigen evaluation [20,21]. This is important for the management of COVID-19 patients, given the relationship between changes in cell number and immunophenotype and disease severity [22]. Additionally, flow cytometry analysis may be useful for better understanding the pathogenesis of COVID-19.

Patients with a severe course of COVID-19 usually require hospitalization, sometimes also in intensive care units (ICUs). In this group of patients, chest CT reveals typical lung lesions that can be useful for screening as well as establishing the diagnosis and selecting appropriate treatment [5,23]. However, there is a significant overlap in imaging results between COVID-19 and other infectious diseases. An additional COVID-19 Reporting and Data System (CORADS) assessment scheme has been shown to categorize chest computed tomography into groups related to lung involvement in COVID-19 patients. This system has been developed and tested in patients with moderate to severe clinical disease [24].

The aim of our study was to determine the leukocyte and lymphocyte populations by flow cytometry methods at selected time points of hospitalization and during the convalescence period, and lung involvement, considering the severity of the disease.

## 2. Materials and Methods

### 2.1. Patients

A total of 46 patients infected by SARS-CoV-2 were determined to be positive based on RT-PCR assays of nasopharyngeal swab specimens according to the WHO guidelines. Patients positive for SARS-CoV-2 were recruited from January 2022 to December 2022 at the Department of Internal Medicine, Pulmonology, Allergology and Clinical Immunology. There were 22 women and 24 men with a mean age of 63 (Q1–Q3: 53–69) years. Patients were divided into two groups based on the need for mechanical ventilation and high-flow oxygen therapy. Patients were divided into non-severe (i.e., conventional oxygen therapy up to 15 L/min) and severe (>15 L/min: mechanical ventilation, NIV, HFNOT). The first group with a severe course of the disease consisted of 16 patients—the severe COVID-19 group. The second group comprised 30 patients without a severe course—the non-severe COVID-19 group with no oxygen administration. Not all patients participated in the subsequent stages of the study (T0–T1–T2). The possibility of obtaining material was limited for logistical, medical and time reasons; therefore, the number of patients varies at each diagnostic stage. Only 7 patients participated in all stages of the study (T0, T1, T2). Computed tomography (CT) imaging was used for the detection of lung abnormalities. Serial chest CT imaging with different time intervals is also effective in estimating the evolution of the disease [25]. Chest CT images of patients were categorized according to the CT chest score system CSS and CORADS classifications [26].

The treatment procedure for COVID-19 patients was carried out in accordance with current knowledge and recommendations of the Polish Society of Epidemiologists and Infectiologists [27]. The average length of stay in hospital was 20 days (Q1–Q3: 11–31).

All patients gave informed consent to participate in the study (Bioethics Committee at the Military Institute of Medicine No. 25/WIM/2021 of 16 June 2021).

### 2.2. Peripheral Blood Samples

The routine test for white blood cell count (WBC) was performed using a hematological analyzer, the Sysmex XN-1500 (Sysmex Corp., Kobe, Japan).

### 2.3. Flow Cytometry

The immunophenotyping test was carried out on whole blood. Leukocyte and lymphocyte subsets were measured by multiparameter flow cytometry method with a panel of monoclonal antibodies using a DxFlex flow cytometer (Beckman Coulter Life Science, Brea, CA, HQ, United States). For detection of surface markers on leukocytes and lymphocytes T, B, NK were stained with fluorescently labelled antibodies: CD4-FITC (catalog number: 345768, clone number: SK3), CD56-PE (catalog number: 345810, clone number: MY31), CD3-PerCP-Cy5.5 (catalog number: 332771, clone number: SK7), CD19-PE-Cy7 (catalog number: 341113, clone number: SJ25C1), CD8-APC (catalog number: 345775, clone number: SK1), CD16-APC-H7 (catalog number: 560195, clone number: 3G8), HLA-DR-V450 (catalog number: 655874, clone number: L243) and CD45-V500 (catalog number: 655873, clone number: 2D1), (BD Bioscience, Becton Dickinson, Franklin Lakes, NJ, USA).

Samples were incubated for 20 min at room temperature. The erythrocytes were lysed with BD Pharm LyseTM Lysing Buffer (Cat. No. 555899). After two washings, cells were analyzed within 2 h. For each sample, a minimum of 20,000 events were collected (white blood cells).

We distinguished the following subpopulations of leukocytes and lymphocytes:

Lymphocytes CD45+ bright: CD45+

Lymphocytes T: CD45+ bright CD3+

Lymphocytes T helper: CD45+ bright CD3+ CD4+

Lymphocytes T cytotoxic: CD45+ bright CD3+ CD8+

Lymphocytes B: CD45+ bright CD19+

Lymphocytes NK: CD45+ bright CD16+ CD3-

Neutrophils: CD45+ CD16+ SSC+

Eosinophils: CD45+ bright SSC+

Basophils: CD45+

Monocytes: CD45+ HLA-DR+

Monocytes activated: CD45+ HLA-DR+ CD16+

The representative gating strategy of PB cells with antibodies specific for leukocytes and lymphocytes subpopulations is presented in Figure 1.

Data were analyzed with Cytexpert and Kaluza C version1.1 (Beckman Coulter Life Science, Brea, CA, HQ, USA), and Infinicyt 1.8 Flow Cytometry (Cytognos, Salamanca, Spain).

### 2.4. Statistical Analysis

The distribution of continuous variables was presented as the median and interquartile range (IQR) in the case of descriptive statistics, while in comparisons as the mean and standard deviation (SD). In all analyses performed, *p* < 0.05 was considered statistically significant. The non-parametric Mann–Whitney *U* test was used to compare continuous variables. To assess the relationship between two continuous variables, correlation calculations were performed, and the Ch-Square test of independence was used to test for a relationship between two dichotomous variables. In order to assess the significance of changes in individual parameters, the Student’s *t*-test was used for related variables or, in the case of dichotomous variables, the McNemar test.

Changes in flow cytometry results at three time points of hospitalization and follow-up were assessed using Friedman’s test (non-parametric test).

In Tables, the following were used: T0—results obtained during hospitalization, T1—results obtained during a follow-up visit 3 months after hospitalization, and T2—results obtained during a follow-up visit 6 months after hospitalization.

Calculations were performed in the R program (R studio 2022.02.2, packages readxl, dplyr, ggpubr, ggplot2, reshape2, corrplot, corrr, Hmisc, openxlsx, writexl) and in Excel (Microsoft Office 365).

## 3. Results

### 3.1. Clinical Characteristics of Patients

A summary of clinical characteristics of the patients and basic data on the course of hospitalization is presented in the Table below (Table 1).

The most common COVID-19 symptoms in hospitalized patients were weakness (93%), fever (80%) and dyspnea (72%). At least half of the patients also experienced cough (63%), myalgia (63%) and headache (50%).

In addition, we presented the clinical characteristics of the patients according to the severity of the disease (Table 2). The most common symptoms in patients with a severe course of COVID-19 were weakness (100%), fever (81%) and shortness of breath (75%). At least half of the patients also experienced cough (63%), myalgia (63%) and shivering (50%).

The most common symptoms occurring in patients without a severe course of COVID-19 were weakness (90%), fever (80%) and shortness of breath (70%). At least half of the patients also experienced cough (63%), myalgia (63%) and headache (53%).

### 3.2. Change in Flow Cytometry Results at Three Time Points of Hospitalization and Follow-Up

For seven patients (none of them had a severe COVID-19 course) the results at three time points were available. Statistically significantly different results in individual measurements (T0–T1–T2) were obtained for the following variables: CD8 T lymphocytes (respectively, median: 3.8 vs. 8.6 vs. 9.8%, *p* = 0.0498), NK cells (respectively, 2.7 vs. 3.5 vs. 4.6%, *p* = 0.002), neutrophils (71.8 vs. 55.2 vs. 45.4%, *p* = 0.028), eosinophils (0.2 vs. 1.8 vs. 1.4, *p* = 0.005), T lymphocytes (as % of lymphocytes, 70.2 vs. 81.0 vs. 77.9, *p* = 0.0498), CD8+ T lymphocytes (as a % of lymphocytes, 16.3 vs. 23.1 vs. 19.3%, *p* = 0.002), B lymphocytes (as a % of lymphocytes, 16.5 vs. 9.0 vs. 9.5, *p* = 0.028), CD8+ T lymphocytes (k/µL, 317.7 vs. 391.2 vs. 542.9, *p* = 0.028), NK cells ((k/µL, 190.6 vs. 192.3 vs. 328.2, *p* = 0.018) (Table 3).

For comparison, Table 4 presents medians and interquartile ranges for the entire population at three time points of hospitalization and follow-up. For various reasons, not every patient was subjected to cytometric tests at specific time points, hence the differences in the number of patients in individual time groups. However, there was a subsequent increase in T cells and eosinophils during hospitalization.

### 3.3. Flow Cytometry Results Depending on the Severity of the Disease

#### 3.3.1. Comparison of Cytometry Results at the Time of Hospitalization (T0)

Comparing patients at the time of hospitalization (T0) by disease severity, we observed a statistically significantly lower percentage of lymphocytes and T lymphocytes in patients with severe disease compared to patients without severe disease (respectively, 25.5 vs. 38.0%, *p* = 0.021 and 18.5 vs. 30.8%, *p* = 0.007) (Table 5). There was a lower percentage of CD4+ T cells and CD8+ cells in severe patients compared to non-severe patients (respectively, 12.1 vs. 18.4, *p* = 0.012 and 5.7 vs. 11.4, *p* = 0.023). There was a higher percentage of neutrophils in severe patients compared to non-severe patients (respectively, 66.8 vs. 51.3%, *p* = 0.012). Lower percentages of eosinophils and basophils were observed in severe patients compared to non-severe patients (respectively, 0.8 vs. 3.1%, *p* = 0.001 and 0.4 vs. 1.0, *p* = 0.016). There was a lower percentage of monocytes with CD16+ expression in severe patients compared to non-severe patients (respectively, 9.7 vs. 18.0%, *p* = 0.005).

#### 3.3.2. Comparison of Cytometry Results Obtained 3 Months after Hospitalization (T1)

Comparing patients by disease severity at 3 months after hospitalization (T1), we observed a statistically significantly higher percentage of basophils in patients with severe disease compared to patients without severe disease (respectively, 0.7 vs. 0.4, *p* = 0.010) (Table 6).

#### 3.3.3. Comparison of Cytometry Results Obtained 6 Months after Hospitalization (T2)

Comparing patients by disease severity at 6 months after hospitalization (T2), we observed a statistically significantly higher ratio of CD4:CD8 cells in patients with severe disease compared to patients without severe disease (respectively, 2.0 vs. 1.4, *p* = 0.035) (Table 7). There was a higher percentage of lymphocytes (respectively, among all cells and as a % of lymphocytes) in severe patients compared to non-severe patients (respectively, 3.8 vs. 2.0, *p* = 0.031 and 10.2 vs. 6.3%, *p* = 0.017).

### 3.4. Comparison of Flow Cytometry Results Depending on the Number of Days of Hospitalization

The number of days of hospitalization was positively correlated with the following cytometry results during hospitalization: T lymphocytes (r = 0.45, *p* = 0.022), CD8 T lymphocytes (0.45, *p* = 0.024) and T lymphocytes (as % of lymphocytes) (r = 0.53, *p* = 007). There was also a negative correlation between the number of days of hospitalization and the percentage of neutrophils (not statistically significant, r = −0.39, *p* = 0.053) (Table 8).

### 3.5. Descriptive Statistics of Radiological Parameters at Different Time Points of Hospitalization and Follow-Up, and Correlation with Flow Cytometry Results

A detailed summary of the results of radiological examinations is presented in the table below (Table 9). The median CT CSS total score was 15 (IQR: 12–17) at hospitalization, 8.5 (IQR: 5–11) at follow-up 3 months after hospitalization, and 6 (IQR: 4–9) at 6 months after hospitalization. The changes between the results after 3 months and during hospitalization, between the results after 6 months and after 3 months, and between the results after 6 months and during hospitalization were statistically significant (*p* < 0.001, *p* = 0.0499 and *p* < 0.001, respectively) (Figure 2).

During hospitalization, all patients for whom the results of the CT CORADS study were available (N = 45), except for one patient, obtained the CT CORADS result = 5, and one patient obtained the CT CORADS result = 1.

Of the patients who underwent this examination at 3 months after hospitalization (N = 39), 95% of patients had a CT CORADS result of 6, and of those who had this study at 6 months after hospitalization, 100% of patients had a CT CORADS result of = 6.

Deviations from the CT CORADS score set: 5 at the time of hospitalization and 6 at follow-up occurred in only 2 patients: one of them had a score of 1 during hospitalization and after 3 months (no data on the result of the follow-up after 6 months), and the other had a score of 5 during hospitalization, replaced then by a score of 2 during the follow-up visit after 3 months (no data on the result of the follow-up after 6 months).

The percentages of patients for whom radiological symptoms occurred during subsequent examinations are summarized in Table 9.

The correlation between the results of flow cytometry during hospitalization with the CT CSS results at the time of hospitalization (T0) was not statistically significant for any of the assessed parameters.

Among the parameters assessed 3 months after hospitalization, a statistically significant relationship with the CT CSS results was observed only for the B lymphocytes (r = −0.43, *p* = 0.010).

In the case of parameters assessed 6 months after hospitalization, the following were observed: weak to moderate negative correlation for lymphocytes (r = −0.47, *p* = 0.013), T lymphocytes (r = −0.43, *p* = 0.026), CD4 T lymphocytes (r = −0.63, *p* < 0.001), CD4:CD8 ratio parameter (r = −0.49, *p* = 0.010), B lymphocytes (r = −0.60, *p* = 0.001), eosinophils (r = −0.41, *p* = 0.036), CD4 T lymphocytes (as % of lymphocytes, r = −0.47, *p* = 0.012), B lymphocytes (as % of lymphocytes, r = −0.58, *p* = 0.002), T CD4 lymphocytes (as k/µL, r = −0.47, *p* = 0.012) and B lymphocytes (as k/µL, r = −0.56, *p* = 0.002). Weak to moderate positive correlations were observed with neutrophils (r = 0.46, *p* = 0.015), CD8 T lymphocytes (as % of lymphocytes, r = 0.38, *p* = 0.049), NK cells (as % of lymphocytes, r = 0.60, *p* = 0.001), and NK cells (as k/µL, r = 0.47, *p* = 0.014) (Table 10).

### 3.6. Comparison of Cytometry Results at Different Time Points of Hospitalization and Follow-Up, with the Severity of the Disease

Then, the dynamics of the variability of the examined cytometry parameters over time was discussed. The results were analyzed in subgroups distinguished according to the severity of the course of the disease (severe and non-severe). The Tables show differences in the results during hospitalization and 3 months after hospitalization (T0–T1, Appendix A
Appendix A), results at 3 months and 6 months after hospitalization (T1–T2, Appendix A
Appendix A) and results during hospitalization and 6 months after hospitalization (T0–T2, Appendix A
Appendix A) (Figure 3).

The results during hospitalization (T0) differed statistically significantly from the results at 3 months after hospitalization (T1) in the non-severe COVID-19 population for the following parameters: CD8 T lymphocytes (median: 6.1 vs. 9.1%, *p* = 0.019), CD4/CD8 ratio (2.3 vs. 1.5%, *p* = 0.002), neutrophils (71.0 vs. 55.2%, *p* = 0.041), eosinophils (0.2 vs. 2.0, *p* = 0.002), basophils (0.3 vs. 0.8, *p* = 0.028), T lymphocytes (as % of lymphocytes, 69.7 vs. 80.4, *p* = 0.002), CD8 T lymphocytes (as % of lymphocytes, 23.0 vs. 29.6, *p* < 0.001), B lymphocytes (as % of lymphocytes, 14.6 vs. 90%, *p* = 0.002) and B lymphocytes (as k/µL, 238.8 vs. 134.3 k/µL, *p* = 0.007) (Appendix A
Appendix A) (Figure 3).

The results during hospitalization (T0) differed statistically significantly from the results at 3 months after hospitalization (T1) in the severe COVID-19 population only for the following parameters: CD8 T lymphocytes (as % of lymphocytes, 23.5 vs. 32.5%, *p* = 0.011) and CD4 T lymphocytes (as % of lymphocytes, 51.3 vs. 47.7%, *p* = 0.017) (Figure 3).

The results at 3 months after hospitalization (T1) differed significantly from the results 6 months after hospitalization (T2) only for B lymphocytes (k/µL, 162 vs. 235.5 k/µL, *p* = 0.039) in the population with a non-severe COVID-19 course (Appendix A
Appendix A) (Figure 3).

The results during hospitalization (T0) differed statistically significantly from the results at 6 months after hospitalization (T2) in the non-severe COVID-19 population for the following parameters: WBC (8900 vs. 6110 k/µL, *p* = 0.036), lymphocytes (22.4 vs. 43.9%, *p* = 0.006), T lymphocytes (16.6 vs. 32.5%, *p* = 0.006), CD4 T lymphocytes (10.3 vs. 20.0%, *p* = 0.014), CD8 T lymphocytes (3.8 vs. 8.2%, *p* = 0.002), CD4/CD8 ratio (2.6 vs. 2.0%, *p* = 0.009), NK cells (2.7 vs. 4.1%, *p* = 0.030), neutrophils (70.2 vs. 49.1%, *p* = 0.003), eosinophils (0.2 vs. 1.7%, *p* < 0.001), T lymphocytes (as % of lymphocytes, 70.2 vs. 77.9%, *p* = 0.016), CD8 T lymphocytes (as % of lymphocytes, 16.3 vs. 25.8, *p* < 0.001), B lymphocytes (as % of lymphocytes, 16.3 vs. 9.7%, *p* = 0.006) and CD8 T lymphocytes (as k/µL, 332.9 vs. 526.1 k/µL, *p* = 0.020) (Appendix A
Appendix A) (Figure 4).

It is noteworthy that the size of the subgroup of patients with a severe course, in whom the appropriate pair of measurements were made during the time points compared, was very low (combination of hospitalization with the first follow-up visit: seven patients, first vs. second follow-up visit: three patients, hospitalization vs. second follow-up visit: two patients). Comparisons for these subgroups showed no statistically significant differences between cytometric parameters at the given time points. The exceptions were statistically significant changes in the parameters CD4 lymphocytes (as a % of lymphocytes) and CD8 lymphocytes (as a % of lymphocytes)between the results obtained at the follow-up visit 3 months after hospitalization (T1) and the results obtained during hospitalization (T0).

## 4. Discussion

Most patients infected with SARS-CoV-2 have mild symptoms, or the disease is asymptomatic. Only people in an at-risk group will develop severe symptoms, with changes in the lungs requiring hospitalization and oxygen administration [26,28].

Research on the immune system is still ongoing in people with COVID-19 after the disease. A comprehensive assessment of the subpopulation of leukocytes and lymphocytes in a group of patients with severe and non-severe courses at selected time points was carried out, and the correlation between the severity of CT lesions (CT CSS) and the results of cytometry was examined. Detection and identification of these populations by flow cytometry may contribute to better understanding the immune system after SARS-CoV-2 infection. In addition, this study will help to clarify long-term symptoms in people cured of COVID-19.

Most of the patients examined in our study showed characteristic clinical symptoms of COVID-19: fever, cough, dyspnea, weakness, headache and sore throat. Due to acute respiratory failure, eight patients required mechanical ventilation (Table 1). In order to distinguish people with different degrees of disease severity, patients were divided into two groups, taking into account the use of mechanical ventilation and high-flow oxygen therapy (Table 2).

We showed changes in leukocyte and lymphocyte subpopulations during the disease at selected time points. Due to the difficulty in collecting data from all patients at chosen points, the results are only available for seven patients, from the non-severe group. Unfortunately, this is a considerable limit of this research; however, even for such a small group, significant changes in some cell populations were perfectly visible. It is noteworthy that there was a significant increase, during hospitalization, of CD8+ T cells, NK cells and eosinophils, with a significant decreasing percentage of neutrophils. Such changes were not observed when all available patients were compared, regardless of disease severity. In COVID-19 patients, the CD8+ T cell population undergoes quantitative and qualitative changes. Reduced cell counts and phenotypes with markers of activation are often observed, especially in severe disease [29]. Other studies also confirmed a decrease in the absolute number of NK cells and profound lymphopenia in patients with COVID-19 [30]. Eosinopenia was one of many factors that may indicate the diagnosis of SARS-CoV-2 infection and the prognosis of a severe course of COVID-19. However, these findings were not pathognomonic for COVID-19 [31].

Dividing and then comparing patients according to the severity of the disease allowed for finding a few differences. At the time of hospitalization, we noticed lower levels of both CD4 and CD8 T lymphocytes, lower rates of eosinophils, basophils, and activated CD16+ monocytes, and elevated neutrophil levels in severe patients relative to non-severe patients. At the next time point, 3 months after hospitalization, we observed higher levels of basophils, and after 6 months, higher CD4/CD8 lymphocytes ratios and T cell levels in the severe compared to non-severe group. The severe course of COVID-19 is associated with higher levels of inflammatory markers than non-severe disease, so monitoring these markers can allow early identification and even prediction of disease progression [32]. Our results are in line with our previous research and data in the literature. Changes in peripheral lymphocyte subsets have been associated with the characteristics of the COVID-19 disease course, as well as dynamic changes in the number of monocytes, the ratio of these cell subsets and the activation state in acute and convalescent COVID-19 patients. Lymphopenia, neutrophilia and reduced levels of eosinophils and basophils are characteristic indicators of SARS-CoV-2 infection [5,21,33,34].

During the first control visit, performed 3 months after infection, the difference in basophil population between the two groups was still significant. A period of 6 months after infection seemed sufficient for COVID-19 patients to obtain results indicating stabilization of the immune system. The accuracy of the choice of time points was confirmed by the results obtained in the CT examination, in which we showed a significant stabilization of changes in the lungs after 3 and 6 months. CT scans can help determine individual patient management and assess the severity of the disease and complications [5]. The CT CSS score classification used for COVID-19 patients determined changes that proved useful in the initial evaluation of patients, not only because of the rapid results, but also because of the ability to assess the initial stage of the disease [26]. The demonstrated correlations in this study between the level of lymphocytes and neutrophils and the time of hospitalization indicated the recovery process of patients.

Next, we showed the dynamics of variability of measured cytometric parameters during the course of COVID-19. The results were analyzed in groups distinguished for disease severity. Patients with non-severe COVID-19 showed significant changes in the leukocyte populations studied at time of hospitalization and at both 3 months and 6 months after onset. We found a significant increase in all studied lymphocyte subpopulations and eosinophils, along with a decrease in neutrophil levels. Three months after hospitalization, in the severe group of patients, we observed a significant difference only in the T cell population. The above studies indicate mainly T cell subpopulations as having the highest dynamics of changes over time and depending on the clinical condition and severity of the disease. Others also found that subsets of T cells showed the greatest significant changes in the course of COVID-19. Gil-Manso S.et al. observed that 10 months after infection, recovered COVID-19 patients showed changes in values of subsets of T and B lymphocytes and innate immune response cells compared to healthy controls [35]. In some studies, multivariate analysis has shown that CD4 and/or CD8 T cell counts independently predict patient outcomes [16,36].

## 5. Conclusions

There is an urgent need for identification of clinical and laboratory predictors for progression in severe and non-severe forms of COVID-19. Our results indicate the need to select groups of patients depending on the stage and clinical symptoms at significant time points that may indicate the direction of further research in understanding the pathogenesis of SARS-CoV-2 infection. We have shown that the results of the CT point system in correlation with cytometric parameters can be used to stratify patients according to the severity of the disease and the assessment of the recovery process. The presented results also point to the usefulness of the multicolor flow cytometry method to assess the recovery of patients with a non-severe course of COVID-19. Patients with severe symptoms of infection require a longer time to normalize clinical parameters. Examination over a period of more than 6 months would be advisable for these patients.

The multitude of data generated by multiparametric flow cytometry, in individual studies and meta-analyses, combined with clinical observations, gives an opportunity to understand better the patient’s immune status and prognostic identification. The use of multicolored flow cytometry in COVID-19 patients with varying degrees of disease severity may be an excellent, simple method for understanding and predicting recovery or disease progression.

## Figures and Tables

**Figure 1 biomedicines-11-02728-f001:**
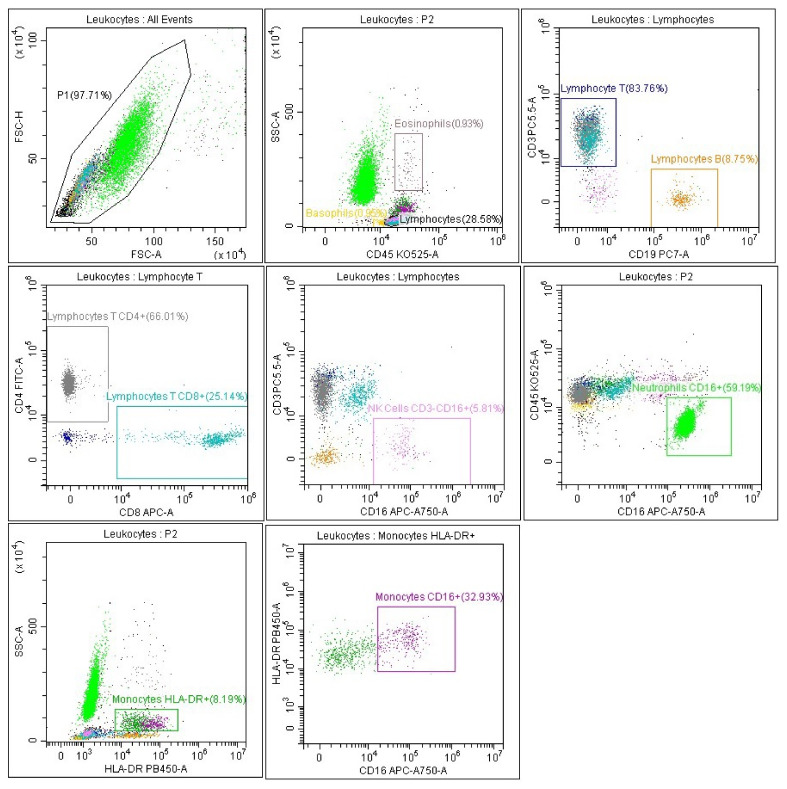
Representative flow cytometry analysis of PB cells with antibodies specific for leukocyte subpopulations. Lymphocyte gating strategy: FSC-A vs. FSC-H plot: For removing clumps and debris, the cells with an equal area and height were gated (greater FSC-A relative to FSC-H) and (very low FSC), CD45 vs. SSC-A plot: Indication of lymphocytes based on their SSC/CD45+ characteristics. Gating strategy for lymphocyte subpopulation: CD3 vs. CD19 plot: T lymphocytes (turquoise and gray) with CD3+ antigen expression and B lymphocytes (yellow) with CD19+ antigen expression. CD4 vs. CD8 plot: T lymphocyte subsets: CD4+ (gray) and CD8+ (turquoise) based on their CD4/CD8 expression. CD3 vs. CD16 plot: NK cells (pink) with CD16+ antigen expression and no expression of CD3 antigen. Neutrophil gating strategy: CD45 vs. CD16 plot: Selection of neutrophils (green) based on their CD16+ high properties and CD45 positive. Monocyte gating strategy: SSC-A vs. HLA-DR plot: Selection of monocytes (dark green) based on their CD45+ high and HLA-DR+ properties. Monocytes with CD16+ expression gating strategy: HLA-DR vs. CD16 plot from the monocyte field: monocytes with CD16+ expression (purple).

**Figure 2 biomedicines-11-02728-f002:**
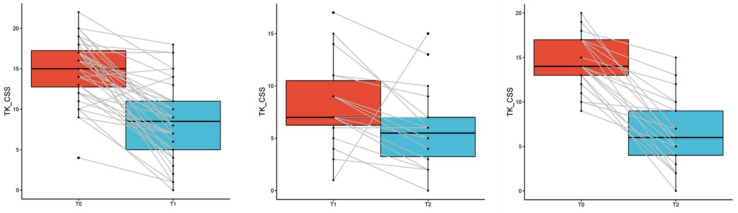
Change in TK CSS intensity between time of hospitalization (T0) and visit 3 months after hospitalization (T1) and visit 6 months after hospitalization (T2).

**Figure 3 biomedicines-11-02728-f003:**
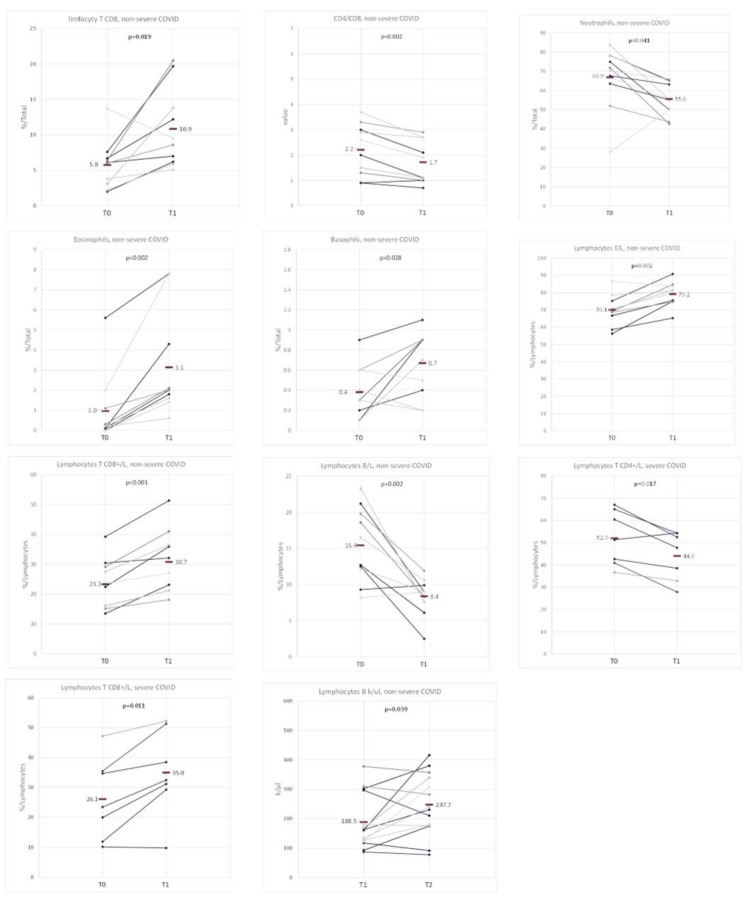
Comparison of cytometry results at different time points of hospitalization and follow-up (T0–T1 and T1–T2) and the severity of the disease (non-severe and severe). The graphs show those variables for which a statistically significant difference was obtained between the distinguished measurements in any of the groups: in the population of patients without severe COVID-19 and the population of patients with severe COVID-19. The average values are marked with a horizontal line.

**Figure 4 biomedicines-11-02728-f004:**
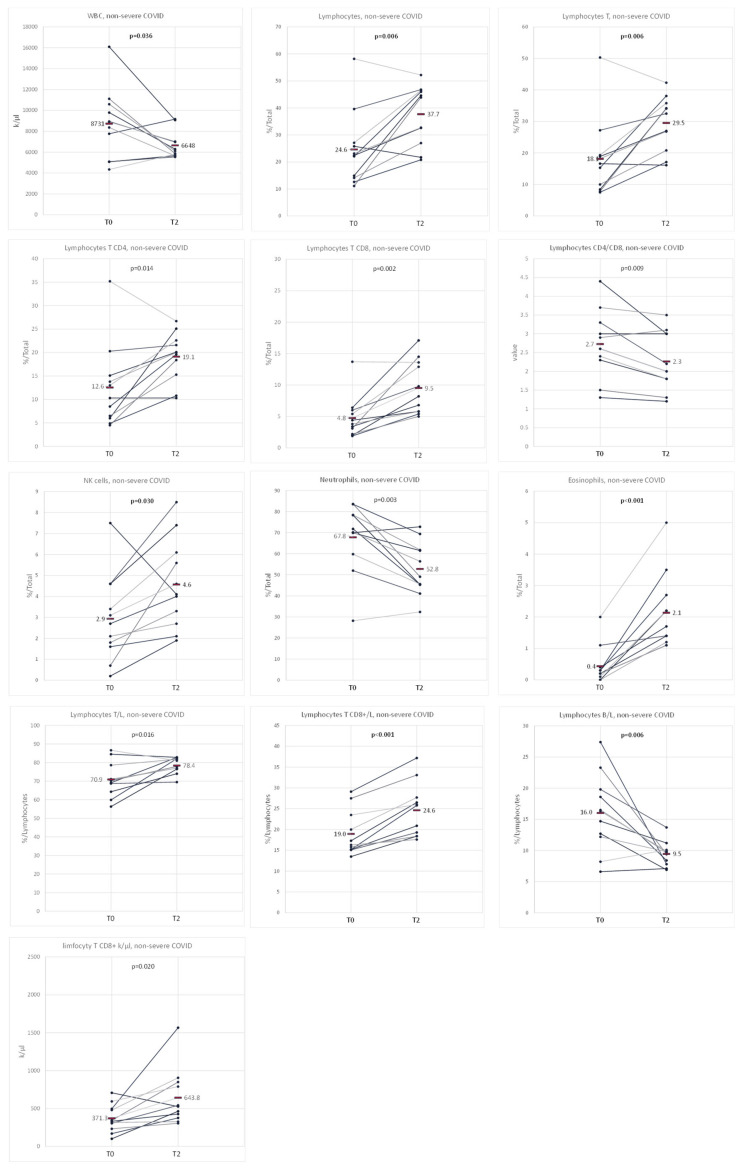
Comparison of cytometry results at different time points of hospitalization and follow-up (T0–T2) and non-severe course of COVID-19. The graphs show those variables for which a statistically significant difference was obtained between the distinguished measurements in the population of patients with a non-severe COVID-19 course. The average values are marked with a horizontal line.

**Table 1 biomedicines-11-02728-t001:** Patient characteristics.

Parameter	Total Patients (n = 46)
Median age, IQR	63 (53–69)
Sex	
▪ Men	24 (52%)
▪ Women	22 (48%)
Smoker (current or past)	8 (17%)
Comorbidities	
▪ COPD	4 (9%)
▪ Asthma	8 (17%)
▪ Other lung diseases	3 (7%)
▪ Heart failure	3 (7%)
▪ Chronic ischemic heart disease (CHD)	7 (15%)
▪ Hypertension	25 (54%)
▪ Diabetes	11 (24%)
▪ Obesity	17 (37%)
▪ Cancers	0 (0%)
▪ Immune deficiencies	0 (0%)
Length of hospitalization (n = 45) median, IQR	20 (11–31)
Oxygen support	
▪ Mechanical ventilation	8 (17%)
▪ NIV	4 (9%)
▪ HFNOT	14 (30%)
▪ Passive oxygen therapy	44 (96%)
Applied treatment	
▪ GKS	37 (80%)
▪ Remdesivir	8 (17%)
▪ Tocilizumab	5 (11%)
▪ Plasma	17 (37%)
COVID-19 symptoms (can coexist)	
▪ Fever	37 (80%)
▪ Cough	29 (63%)
▪ Dyspnea	33 (72%)
▪ Weakness	43 (93%)
▪ Myalgia	29 (63%)
▪ Shivering	20 (43%)
▪ Headaches	23 (50%)
▪ Diarrhea	5 (11%)
▪ Nausea	5 (11%)
▪ Sore throat	7 (15%)
▪ Disorders in/s	18 (39%)

**Table 2 biomedicines-11-02728-t002:** Patient clinical characteristics depending on the severity of the disease course.

Parameter	Patients with Severe COVID-19 (n = 16)	Without a Severe Course of COVID-19 (n = 30)	*p*-Value
Median age, IQR	64 (57–66)	63 (57–72)	0.835 (Mann-Whitney U test)
Sex			
▪ men	6 (38%)	18 (60%)	0.146
▪ women	10 (63%)	12 (40%)	0.146
Smoker (current or past)	2 (13%)	6 (20%)	0.523
Comorbidities			
▪ COPD	1 (6%)	3 (10%)	0.667
▪ Asthma	4 (25%)	4 (13%)	0.320
▪ Other lung diseases	2 (13%)	1 (3%)	0.230
▪ NS (Heart failure?)	1 (6%)	2 (7%)	0.957
▪ Chronic ischemic heart disease (CHD)	0 (0%)	7 (23%)	0.036
▪ Hypertension	9 (56%)	16 (53%)	0.850
▪ Diabetes	4 (25%)	7 (23%)	0.900
▪ Obesity	8 (50%)	9 (30%)	0.181
▪ Cancers	0 (0%)	0 (0%)	-
▪ Immune deficiencies	0 (0%)	0 (0%)	-
Length of hospitalization median, IQR	32 (24–48)	12 (10–20)	<0.001 (Mann-Whitney U test)
Oxygen support			
▪ Mechanical ventilation	8 (50%)	0 (0%)	-
▪ NIV	4 (25%)	0 (0%)	-
▪ HFNOT	14 (88%)	0 (0%)	-
▪ Passive oxygen therapy	16 (100%)	28 (93%)	0.291
Applied treatment			
▪ GKS	16 (100%)	21 (70%)	0.015
▪ Remdesivir	3 (19%)	5 (17%)	0.859
▪ Tocilizumab	4 (25%)	1 (3%)	0.025
▪ Plasma	6 (38%)	611 (37%)	0.956
COVID-19 symptoms (can coexist)			
▪ Fever	13 (81%)	24 (80%)	0.919
▪ Cough	10 (63%)	19 (63%)	0.956
▪ Dyspnea	12 (75%)	21 (70%)	0.720
▪ Weakness	16 (100%)	27 (90%)	0.191
▪ Muscle aches	10 (63%)	19 (63%)	0.956
▪ Chills	8 (50%)	12 (40%)	0.515
▪ Headaches	7 (44%)	16 (53%)	0.536
▪ Diarrhea	2 (13%)	3 (10%)	0.795
▪ Nausea	2 (13%)	3 (10%)	0.795
▪ Sore throat	1 (6%)	6 (20%)	0.216
▪ Disorders in/s	6 (38%)	12 (40%)	0.869

**Table 3 biomedicines-11-02728-t003:** Flow cytometry results at three time points of hospitalization and follow-up (n = 7): T0—time of hospitalization; T1—3 months after hospitalization; T2—6 months after hospitalization. * *p* < 0.05.

Parameter	T0 Median (Q1–Q3)	T1 Median (Q1–Q3)	T2 Median (Q1–Q3)	*p*-Value
WBC	7760 (5080–9475)	5770 (5080–6215)	5810 (5650–5985)	0.965
lymphocytes (%)	22.1 (14.5–31.4)	35.1 (27.2–42.6)	44.6 (38.3–46.4)	0.066
lymphocytes_T (%)	15.3 (9.2–22.7)	29.5 (21.1–33.1)	34.1 (29.7–36.2)	0.066
lymphocytes T CD4 (%)	8.5 (6.3–17.1)	15.8 (14.9–19.6)	20 (19–23.4)	0.066
lymphocytes T CD8 (%)	3.8 (2.7–6.2)	8.6 (6.1–11.7)	9.8 (7–14.1)	* 0.0498
CD4:CD8	2.9 (2.1–3.2)	2.1 (1.5–2.7)	2.2 (1.7–3.1)	0.090
cells_NK (%)	2.7 (2–3.9)	3.5 (2.7–4.6)	4.6 (3.7–6.5)	* 0.002
lymphocytes_B (%)	2.8 (2.5–4.5)	2.9 (2.5–3.6)	3.9 (3.2–4.6)	0.331
neutrophils (%)	71.8 (61.1–78.4)	55.2 (46.9–65)	45.4 (43.2–52.8)	* 0.028
eosinophils (%)	0.2 (0.1–0.7)	1.8 (1.5–2.1)	1.4 (1.2–2.9)	* 0.005
basophils (%)	0.3 (0.3–0.5)	0.7 (0.4–0.9)	0.7 (0.6–0.9)	0.331
monocytes (%)	6.5 (6.4–7.2)	6.1 (5.2–6.8)	7 (5.3–9.2)	0.156
monocytes CD16+ (%)	6.9 (5.7–15.2)	13.6 (10.4–19.9)	10.3 (6.2–10.7)	0.368
lymphocytes_T (%L)	70.2 (68.9–74.7)	81 (77.4–82.7)	77.9 (76.8–81.4)	* 0.0498
lymphocytes T CD4 (%L)	45.8 (40.3–55.6)	52.9 (45.3–54.7)	51.2 (46.3–56.4)	0.331
lymphocytes T CD8 (%L)	16.3 (15.5–25.5)	23.1 (21.2–31.8)	19.3 (18.4–29.6)	* 0.002
cells_NK (%L)	11.5 (7.9–12.6)	10 (8.5–12.2)	12.7 (8.9–14.6)	0.651
lymphocytes_B (%L)	16.5 (12.5–19.2)	9 (8–9.8)	9.7 (8.9–10)	* 0.028
lymphocytes T (k/µL)	1187.3 (962.8–1453)	1427.6 (1162.6–1710.4)	1930.1 (1656–2230.9)	0.156
lymphocytes T CD4 (k/µL)	688.4 (579.8–1093.5)	802 (745.3–1216.3)	1196.6 (1103.1–1486)	0.156
lymphocytes T CD8_(k/µL)	317.7 (269.2–420.5)	391.2 (367.4–557)	542.9 (395.6–819.9)	* 0.028
NK cells (k/µL)	190.6 (155.1–221.4)	192.3 (164–306.4)	328.2 (234.4–392.6)	* 0.018
lymphocytes B (k/µL)	243.6 (221.2–303.5)	134 (124.7–244.6)	240.3 (178.8–295.2)	0.276

**Table 4 biomedicines-11-02728-t004:** Medians and interquartile ranges at three time points of hospitalization and follow-up: T0—time of hospitalization; T1—3 months after hospitalization; T2—6 months after hospitalization.

Parameter	T0 Median (Q1–Q3) n = 25	T1 Median (Q1–Q3) n = 34	T2 Median (Q1–Q3) n = 27
WBC	7760 (5070–9480)	5830 (4807.5–6682.5)	6290 (5735–7435)
lymphocytes (%)	30 (20.0–39.5)	31.9 (26.7–39.2)	36.7 (26.2–45.3)
lymphocytes_T (%)	20 (14.5–30.9)	24.2 (20.2–34.1)	27.6 (18.9–35.0)
lymphocytes T CD4 (%)	15.1 (6.7–19.2)	14.6 (12–19.9)	18.7 (11–22.1)
lymphocytes T CD8 (%)	6.1 (3.8–11.8)	8.9 (7–12.1)	8.7 (6.2–13.4)
CD4:CD8	2.3 (1.3–2.9)	1.8 (1.1–2.1)	1.7 (1.3–2.2)
cells NK (%)	3.1 (1.8–4.3)	3.8 (2.6–5.2)	3.3 (2.6–4.7)
lymphocytes_B (%)	3.5 (2.5–4.8)	2.8 (1.9–3.2)	3.3 (1.6–4.6)
neutrophils (%)	59.8 (50.8–71.8)	57.7 (48.6–64.3)	56.4 (45.8–62.3)
eosinophils (%)	0.8 (0.2–2.9)	2.6 (1.7–3.8)	1.7 (1.2–2.4)
basophils (%)	0.4 (0.3–1.1)	0.6 (0.4–0.9)	0.6 (0.2–1.0)
monocytes (%)	6.5 (4.9–7.8)	6.3 (5.4–7.0)	6 (4.5–8.6)
monocytes CD16+ (%)	11.2 (6.9–18.5)	15.4 (8.3–20.4)	10.5 (5.6–15.7)
lymphocytes T (%L)	75.4 (69.1–80.5)	80.2 (75.1–84.5)	79.4 (74.9–82.8)
lymphocytes T CD4 (%L)	43.6 (39.8–51.3)	48.4 (39.9–53.1)	48.7 (43.2–51.8)
lymphocytes T CD8 (%L)	23.5 (15.8–31.5)	27.6 (22–35.7)	27.6 (22.4–34.2)
cells NK (%L)	11.6 (6.9–12.7)	11.1 (9–15.5)	10.7 (7.6–14.6)
lymphocytes_B (%L)	12.7 (9.5–16.3)	8.6 (6.7–10.4)	8.7 (7.1–9.9)
lymphocytes T (k/µL)	1364.9 (1173.8–1719.9)	1413.9 (1089.9–1911.6)	1757.1 (1402.4–2045.3)
lymphocytes T CD4 (k/µL)	968 (646.4–1157)	842.9 (673.9–1130.1)	1095.8 (851–1382.5)
lymphocytes T CD8_(k/µL)	480.6 (317.7–667.9)	474.5 (363.5–738.4)	569 (445.9–819.9)
NK cells (k/µL)	197 (134.5–243.4)	214 (133.2–328.5)	247.9 (155.4–357.1)
lymphocytes B (k/µL)	243.6 (185.8–323.1)	162 (95.6–189)	203.7 (129.8–295.2)

**Table 5 biomedicines-11-02728-t005:** Results of cytometry depending on the severity of the COVID course at the time of hospitalization (T0). * *p* < 0.05.

Parameter	n Total	Mean (SD) All Patients	n Severe Course	Mean (SD)—Severe Course	n Non-Severe Course	Mean (SD) Non-Severe Course	*p*-Value
T0							
WBC (k/µL)	25	7662 (3027.3)	10	6410 (2738.2)	15	8496.7 (3004.1)	0.056
lymphocytes (%)	25	30.5 (13.4)	10	38 (11.2)	15	25.5 (12.8)	* 0.021
lymphocytes_T (%)	25	23.4 (12)	10	30.8 (9.6)	15	18.5 (11.1)	* 0.007
lymphocytes T CD4 (%)	25	14.6 (7.6)	10	18.4 (5.2)	15	12.1 (8.1)	* 0.012
lymphocytes TCD8 (%)	25	7.9 (5.8)	10	11.4 (7.1)	15	5.7 (3.5)	* 0.023
CD4:CD8	25	2.4 (1.4)	10	2.4 (1.9)	15	2.4 (1)	0.373
cells_NK (%)	25	3.2 (1.8)	10	3.1 (1.4)	15	3.3 (2.1)	0.846
lymphocytes_B (%)	25	3.9 (1.7)	10	4.1 (1.6)	15	3.7 (1.8)	0.505
neutrophils (%)	25	60.6 (15.6)	10	51.3 (12.3)	15	66.8 (14.6)	* 0.012
eosinophils (%)	25	1.7 (2.1)	10	3.1 (2.2)	15	0.8 (1.4)	* 0.001
basophils (%)	25	0.6 (0.5)	10	1.0 (0.6)	15	0.4 (0.3)	* 0.016
monocytes (%)	25	6.7 (2.3)	10	6.6 (2.3)	15	6.8 (2.3)	0.890
monocytes CD16+ (%)	25	13 (7.7)	10	18 (6.7)	15	9.7 (6.6)	* 0.005
results in the lymphocyte field (% of lymphocytes)							
lymphocytes T (%L)	25	74.5 (8.8)	10	80.9 (3.5)	15	70.2 (8.7)	* 0.003
lymphocytes T CD4 (%L)	25	46.8 (11)	10	49.2 (11.3)	15	45.2 (10.8)	0.405
lymphocytes T CD8 (%L)	25	24.7 (10.1)	10	29 (12.2)	15	21.9 (7.6)	0.127
cells_NK (%L)	25	11.8 (7.9)	10	8.3 (3.4)	15	14.2 (9.2)	0.056
lymphocytes_B (%L)	25	13.6 (5.3)	10	10.8 (3)	15	15.5 (5.8)	* 0.033
results for lymphocyte subpopulation (k/µL)							
lymphocytes T (k/µL)	25	1657.6 (957.8)	10	2032.4 (1292.5)	15	1407.7 (575.6)	0.305
lymphocytes T CD4 (k/µL)	25	1037.2 (624.1)	10	1230.9 (830.8)	15	908.1 (423.4)	0.488
lymphocytes T CD8_(k/µL)	25	552.8 (438.6)	10	730.9 (628.5)	15	434.2 (195.3)	0.279
NK cells (k/µL)	25	244.9 (232.6)	10	180.7 (92.2)	15	287.7 (287)	0.255
lymphocytes B (k/µL)	25	285.2 (154.8)	10	287.9 (227.4)	15	283.4 (88.5)	0.279

**Table 6 biomedicines-11-02728-t006:** Results of cytometry depending on the severity of the COVID course at 3 months after hospitalization (T1). * *p* < 0.05.

Parameter	n Total	Mean (SD) All Patients	n Severe Course	Mean (SD)—Severe Course	n Non-Severe Course	Mean (SD) Non-Severe Course	*p*-Value
T1							
WBC (k/µL)	34	6097.9 (2307.1)	11	6412.7 (3297.9)	23	5947.4 (1722.5)	0.912
lymphocytes (%)	34	33.3 (10.4)	11	30.7 (9.1)	23	34.5 (11)	0.429
lymphocytes_T (%)	34	26.3 (9.5)	11	24.4 (8.8)	23	27.3 (9.9)	0.632
lymphocytes T CD4 (%)	34	15.5 (5.9)	11	14 (6)	23	16.2 (5.9)	0.418
lymphocytes T CD8 (%)	34	10.1 (5.1)	11	9.9 (5.2)	23	10.2 (5.2)	0.768
CD4:CD8	34	1.8 (1)	11	1.8 (1.3)	23	1.8 (0.8)	0.417
cells_NK (%)	34	4.1 (1.7)	11	4 (2.1)	23	4.1 (1.6)	0.782
lymphocytes_B (%)	34	2.8 (1.3)	11	2.2 (0.9)	23	3.1 (1.4)	0.101
neutrophils (%)	34	56.8 (11)	11	59.8 (11.5)	23	55.4 (10.7)	0.357
eosinophils (%)	34	3.2 (2.3)	11	3.1 (1.7)	23	3.3 (2.5)	0.897
basophils (%)	34	0.6 (0.4)	11	0.4 (0.4)	23	0.7 (0.3)	* 0.010
monocytes (%)	34	6.3 (1.4)	11	6.4 (1.6)	23	6.2 (1.3)	0.839
monocytes_CD16+ (%)	34	14.6 (7.8)	11	16 (7.6)	23	14 (8)	0.339
results in the lymphocyte field (% of lymphocytes)							
lymphocytes_T (%L)	34	77.8 (9.9)	11	77.3 (14.4)	23	78.1 (7.2)	0.277
lymphocytes T CD4 (%L)	34	46.1 (9.6)	11	44.1 (12)	23	47 (8.3)	0.686
lymphocytes T CD8 (%L)	34	29.6 (10.2)	11	31.5 (12.9)	23	28.6 (8.9)	0.473
cells_NK (%L)	34	13.3 (6.7)	11	14.5 (9.2)	23	12.7 (5.3)	0.956
lymphocytes_B (%L)	34	8.9 (4.4)	11	8.4 (6.1)	23	9.1 (3.3)	0.204
results for lymphocyte subpopulation (k/µL)							
lymphocytes T (k/µL)	35	1480.1 (704.6)	12	1270.3 (631.3)	23	1589.6 (729)	0.508
lymphocytes T CD4 (k/µL)	34	889.3 (367.6)	11	778 (298)	23	942.5 (391.4)	0.320
lymphocytes T CD8_(k/µL)	34	596.4 (371)	11	579.2 (345.7)	23	604.7 (389.8)	0.971
NK cells (k/µL)	34	252 (142.4)	11	262.5 (187.4)	23	247 (119.9)	0.854
lymphocytes B (k/µL)	34	171.1 (102)	11	153.8 (123.5)	23	179.5 (91.9)	0.462

**Table 7 biomedicines-11-02728-t007:** Results of cytometry depending on the severity of the COVID course at 6 months after hospitalization (T2). * *p* < 0.05.

Parameter	n Total	Mean (SD) All Patients	n Severe Course	Mean (SD)—Severe Course	n Non-Severe Course	Mean (SD) Non-Severe Course	*p*-Value
T2							
WBC (k/µL)	27	6956.3 (2180.6)	7	6901.4 (1924.6)	20	6975.5 (2309.8)	0.978
lymphocytes (%)	27	35.7 (12.2)	7	30.9 (8)	20	37.3 (13.2)	0.158
lymphocytes_T (%)	27	28.3 (10.7)	7	24.8 (7.5)	20	29.5 (11.5)	0.306
lymphocytes T CD4 (%)	27	17.1 (6.1)	7	13.8 (3.9)	20	18.2 (6.4)	0.056
lymphocytes T CD8 (%)	27	10.4 (5.4)	7	10.5 (4.3)	20	10.3 (5.8)	0.599
CD4:CD8	27	1.9 (0.8)	7	1.4 (0.4)	20	2 (0.8)	* 0.035
cells_NK (%)	27	4 (2)	7	4.1 (2.5)	20	4 (1.8)	0.912
lymphocytes_B (%)	27	3.3 (1.7)	7	2 (1.2)	20	3.8 (1.6)	* 0.031
neutrophils (%)	27	55.1 (13.2)	7	59.5 (8.8)	20	53.5 (14.2)	0.234
eosinophils (%)	27	2 (1.2)	7	1.5 (0.7)	20	2.1 (1.3)	0.437
basophils (%)	25	0.6 (0.5)	7	0.6 (0.4)	20	0.7 (0.5)	0.845
monocytes (%)	27	6.5 (2.2)	7	7.3 (2.3)	20	6.2 (2.1)	0.376
monocytes_CD16+ (%)	27	11.1 (7.2)	7	12 (4.9)	20	10.8 (7.9)	0.599
results in the lymphocyte field (% of lymphocytes)							
lymphocytes_T (%L)	27	78.4 (6.5)	7	79.7 (8.2)	20	78 (5.9)	0.391
lymphocytes T CD4 (%L)	27	47.9 (7.6)	7	44.8 (7.7)	20	49 (7.5)	0.256
lymphocytes T CD8 (%L)	27	28.5 (7.9)	7	33.2 (6.8)	20	26.8 (7.7)	0.053
cells_NK (%L)	27	12.4 (6.8)	7	14 (9.1)	20	11.8 (5.9)	0.868
lymphocytes_B (%L)	27	9.2 (4)	7	6.3 (2.7)	20	10.2 (4)	* 0.017
results for lymphocyte subpopulation (k/µL)							
lymphocytes T (k/µL)	27	1886.8 (783.4)	7	1652 (497.3)	20	1969 (856.9)	0.306
lymphocytes T CD4 (k/µL)	27	1126.4 (398.7)	7	918.5 (243.2)	20	1199.1 (421.1)	0.092
lymphocytes T CD8_(k/µL)	27	705.1 (408.8)	7	702.7 (305.7)	20	706 (446.2)	0.599
NK cells (k/µL)	27	272.6 (134.5)	7	276.7 (177.5)	20	271.2 (121.7)	0.846
lymphocytes B (k/µL)	27	216.8 (109.3)	7	134.5 (71.6)	20	245.6 (106.5)	* 0.025

**Table 8 biomedicines-11-02728-t008:** Comparison of flow cytometry results depending on the number of days of hospitalization. * *p* < 0.05.

Parameter	n	r	*p*-Value
WBC	25	−0.21	0.308
lymphocytes (%)	25	0.36	0.075
lymphocytes_T (%)	25	0.45	* 0.022
lymphocytes T CD4 (%)	25	0.35	0.086
lymphocytes T CD8 (%)	25	0.45	* 0.024
CD4:CD8	25	−0.21	0.318
cells_NK (%)	25	−0.08	0.720
lymphocytes_B (%)	25	0.18	0.381
neutrophils (%)	25	−0.39	0.053
eosinophils (%)	25	0.18	0.377
basophils (%)	25	0.25	0.222
monocytes (%)	25	0.14	0.503
monocytes CD16+ (%)	25	0.32	0.120
lymphocytes T (%L)	25	0.53	* 0.007
lymphocytes T CD4 (%L)	25	0.16	0.457
lymphocytes T CD8 (%L)	25	0.35	0.088
cells_NK (%L)	25	−0.35	0.089
lymphocytes_B (%L)	25	−0.30	0.151
lymphocytes T (k/µL)	25	0.35	0.083
lymphocytes T CD4 (k/µL)	25	0.27	0.190
lymphocytes T CD8 (k/µL)	25	0.30	0.151
NK cells (k/µL)	25	−0.14	0.509
lymphocytes B (k/µL)	25	−0.01	0.978

**Table 9 biomedicines-11-02728-t009:** Particular radiological symptoms during various time points of hospitalization and follow-up.

Radiological Examination Results—Hospitalization		Radiological Examination Results—3 Months after Hospitalization		Radiological Examination Results—6 Months after Hospitalization	
CT CCS intensity (n = 46)	15 (12–17)	CT CCS intensity (n = 40)	8.5 (5–11)	CT CCS intensity (n = 21)	6 (4–9)
TK CORADS	N = 45	TK CORADS	N = 39	TK CORADS	N = 46
1	1 (2%)	1	1 (3%)	1	0 (0%)
2	0 (0%)	2	1 (3%)	2	0 (0%)
3	0 (0%)	3	0 (0%)	3	0 (0%)
4	0 (0%)	4	0 (0%)	4	0 (0%)
5	44 (96%)	5	0 (0%)	5	0 (0%)
6	0 (0%)	6	37 (95%)	6	21 (100%)
Matt. Glass (N = 40)	40 (100%)	Matt. Glass (N = 40)	39 (98%)	Matt. Glass (N = 20)	18 (90%)
cobblestone (N = 40)	30 (75%)	cobblestone (N = 40)	3 (8%)	cobblestone (N = 20)	0 (0%)
parenchymal (N = 40)	35 (88%)	parenchymal (N = 40)	8 (20%)	parenchymal (N = 20)	5 (25%)
liquid (N = 40)	3 (8%)	liquid (N = 40)	1 (3%)	liquid (N = 20)	0 (0%)
pneumothorax (N = 41)	1 (2%)	pneumothorax (N = 40)	0 (0%)	pneumothorax (N = 20)	0 (0%)
area hearts (N = 41)	11 (27%)	area hearts (N = 40)	1 (3%)	area hearts (N = 20)	1 (5%)
fiber strands (N = 40)	5 (13%)	fiber strands (N = 40)	33 (83%)	fiber strands (N = 20)	16 (80%)
dilation (N = 40)	33 (83%)	dilation (N = 40)	25 (63%)	dilation (N = 20)	14 (70%)

**Table 10 biomedicines-11-02728-t010:** Correlation of CT CCS results with flow cytometry parameters at different time points of hospitalization and follow-up. * *p* < 0.05.

Parameter	n	r	*p*-Value
T0 cytometry			
WBC	25	−0.21	0.323
lymphocytes (%)	25	0.12	0.583
lymphocytes_T (%)	25	0.16	0.445
lymphocytes T CD4 (%)	25	0.12	0.554
lymphocytes T CD8 (%)	25	0.12	0.560
CD4:CD8	25	−0.14	0.512
cells NK (%)	25	−0.02	0.909
lymphocytes_B (%)	25	0.01	0.957
neutrophils (%)	25	−0.09	0.667
eosinophils (%)	25	0.17	0.427
basophils (%)	25	0.00	0.999
monocytes (%)	25	−0.17	0.429
monocytes CD16+ (%)	25	0.18	0.381
lymphocytes T (%L)	25	0.16	0.446
lymphocytes T CD4 (%L)	25	0.00	0.996
lymphocytes T CD8 (%L)	25	0.13	0.537
cells NK (%L)	25	−0.09	0.667
lymphocytes_B (%L)	25	−0.12	0.569
lymphocytes T (k/µL)	25	−0.07	0.733
lymphocytes T CD4 (k/µL)	25	−0.08	0.708
lymphocytes T CD8 (k/µL)	25	0.03	0.873
NK cells (k/µL)	25	−0.13	0.535
lymphocytes B (k/µL)	25	−0.16	0.448
T1 cytometry			
WBC	34	0.01	0.940
lymphocytes (%)	34	−0.08	0.648
lymphocytes T (%)	34	−0.06	0.722
lymphocytes T CD4 (%)	34	−0.16	0.357
lymphocytes T CD8 (%)	34	0.07	0.684
CD4:CD8	34	−0.11	0.548
cells_NK (%)	34	0.12	0.513
lymphocytes B (%)	34	−0.43	* 0.010
neutrophils (%)	34	0.21	0.241
eosinophils (%)	34	−0.30	0.088
basophils (%)	34	−0.16	0.367
monocytes (%)	34	−0.19	0.273
monocytes CD16+ (%)	34	0.32	0.067
lymphocytes T (%L)	34	0.07	0.708
lymphocytes T CD4 (%L)	34	−0.10	0.568
lymphocytes T CD8 (%L)	34	0.11	0.530
cells_NK (%L)	34	0.21	0.233
lymphocytes B (%L)	34	−0.33	0.053
lymphocytes T (k/µL)	34	−0.01	0.972
lymphocytes T CD4 (k/µL)	34	−0.12	0.499
lymphocytes T CD8 (k/µL)	34	0.06	0.724
NK cells (k/µL)	34	0.13	0.475
lymphocytes B (k/µL)	34	−0.26	0.137
T2 cytometry			
WBC	27	0.29	0.143
lymphocytes (%)	27	−0.47	* 0.013
lymphocytes T (%)	27	−0.43	* 0.026
lymphocytes T CD4 (%)	27	−0.63	<0.001
lymphocytes T CD8 (%)	27	−0.18	0.377
CD4:CD8	27	−0.49	0.010
cells_NK (%)	27	0.29	0.143
lymphocytes B (%)	27	−0.60	* 0.001
neutrophils (%)	27	0.46	* 0.015
eosinophils (%)	27	−0.41	* 0.036
basophils (%)	27	−0.23	0.248
monocytes (%)	27	−0.11	0.569
monocytes CD16+ (%)	27	0.02	0.928
lymphocytes T (%L)	27	−0.30	0.132
lymphocytes T CD4 (%L)	27	−0.47	* 0.012
lymphocytes T CD8 (%L)	27	0.38	* 0.049
NK cells (%L)	27	0.60	* 0.001
lymphocytes B (%L)	27	−0.58	* 0.002
lymphocytes T (k/µL)	27	−0.30	0.128
lymphocytes T CD4 (k/µL)	27	−0.47	0.012
lymphocytes T CD8 (k/µL)	27	0.12	0.567
NK cells (k/µL)	27	0.47	* 0.014
lymphocytes B (k/µL)	27	−0.56	* 0.002

## Data Availability

All relevant data are contained within the article. The original contributions presented in the study are included in the article. Further inquiries can be directed to the corresponding author.

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
