# Peer review of "Analysis of Leukocyte Subpopulations by Flow Cytometry during Hospitalization Depending on the Severity of COVID-19 Course"

_biomedicines, 2023, doi:10.3390/biomedicines11102728_

Round 1

Reviewer 1 Report

This study aims to analyze leukocyte subpopulations using FACS and examine the changes in subcellular types or populations between non-severe and severe COVID-19 patients. The authors attempted to conduct a longitudinal analysis, but due to limitations in the number of enrolled cases, they resorted to a cross-sectional analysis of these two groups at different time points. While their data does provide valuable insights into changes in cell populations or numbers (e.g., CD4 and CD8 T lymphocytes, eosinophils, basophils, and CD16+ monocytes) in severe and non-severe COVID-19 patients during various phases of recovery after SARS-CoV-2 infection, the small sample size raises concerns about result reliability and their clinical significance. Additionally, there are still several points that require further clarification and revision.

Major Points:

 1.In this study, the classification of COVID-19 severity is based on the need for mechanical ventilation and high-flow oxygen therapy. However, the WHO has provided clear criteria for defining severe COVID-19 in its clinical management living guidance. Could the authors consider redefining the patient classifications using these WHO criteria?

 2.The primary aim of the study is to analyze changes in leukocyte subpopulations between severe and non-severe COVID-19 patients at different time points, as presented in Tables 5-7. Unfortunately, only seven non-severe cases were continuously analyzed during these three phases (To-T2). It is unclear how many severe cases were continuously analyzed at two time points, and whether the data revealed any significant differences. To enhance the study's credibility, it is recommended to include age and gender differences in Tables 5-7.

 3.The authors have analyzed clinical syndromes and characteristics between non-severe and severe COVID-19 patients in Table 2, but they have not reported the p-values for these comparisons. It is suggested conducting the analysis and including the statistical significance values.

 4. It is advisable to establish a link between the variation in leukocyte subpopulations and the clinical syndromes, and to analyze potential correlations.

 5. The authors have examined the correlation between CT CORADS scores and changes in leukocyte subpopulations at three time points, which seems unrelated to the study's title. Moreover, it should be noted that CT CORADS is not universally accepted for diagnosing COVID-19 disease progression and is only used in specific countries. Furthermore, it is unclear how image scores can be related to the more sensitive blood cell subpopulation diagnosis.

 6.The authors should provide an explanation for why most tables present both analytical cell population percentages and concentrations in k/μl. Are these values intended to convey different meanings, or is there a specific emphasis the authors wish to make?

 7. Could changes in leukocyte subpopulations at different time points and differences between non-severe and severe COVID-19 patients serve as diagnostic or prognostic biomarkers for the disease? Currently, the authors discuss increases or decreases in certain cell types, but the biological or clinical significance of these variations remains unclear.

8.Given that SARS-CoV-2 variants exhibit varying levels of virulence that correlate with mild or severe symptoms, have the authors considered whether the enrolled patients were infected with different SARS-CoV-2 variants? Has this potential influencing factor been taken into account during the analysis?

 Minor:

 1.      The current placement of Figures and Tables in the text is not conducive to easy review and reading. Please consider reorganizing the positioning of figures and tables within the main text for better readability.

 2.      Some research papers have conducted similar analyses and categorized patients into asymptomatic and moderate COVID-19 groups, suggesting that neutrophils and lymphocytes can help distinguish between these two categories. Is it possible that the non-severe COVID-19 patients in this study include both asymptomatic and moderate cases?

 3.      The resolution of Figures 3 and 4 needs improvement. Many of the lines and text within these figures are too small and unclear. Please enhance the resolution for better visibility and comprehension.

1.       Language editing is necessary for improved clarity and accuracy.

Reviewer 2 Report

The paper summarizes an accountable amount of data. The work needed for that was impressive for sure and I applaud the Authors for that. The amount of presented information also makes the paper difficult to follow at times. I have listed the issues I have seen below, but I need to stress that those mistakes and editorial issues do not reduce the value of the work presented here.

·         In 2.1. section, there is a piece of information that in total 46 patients' blood was analysed. In later parts of the paper are used other numbers of patients and in no time point (T0/T1/T2) 46 patients appear. The Authors have written that due to various reasons, not all patients followed from T0 to T2 time point (and that in itself is understanding) but for clarity's sake please change the description in this section to be more clear.

·         lines 120 - 121

o    were the erythrocytes removed before the flow cytometry analysis?

o    20 000 events is the number of all analysed cells or only WBCs?

·         I suggest moving the Figure 1 into 2.1. section, that way it will be easier to analyse and follow

·         did the statistical data follow normal distribution? Was the Kolmogorov-Smirnov test or Shapiro-Wilk test performed? I am asking because it is unclear why some tests were performed as if there was a normal distribution of data and some as if there was not a normal distribution

·         Table 1. and Table 2. should be moved into 3.1. section;

o    additionally, there should be some editorial changes made in Table 1. to make it more clear --> COVID-19 symptoms should be moved to the left

o    the denotation that all those listed characteristics do not sum up to 100% because they can co-exist should be added in both table 1 and table 2

·         Tables 3. and 4. should be moved into 3.2. section; Tables 5 - 7 should be moved into their respective sections also. It is difficult to follow the analysed data when the reader needs to skip between parts of the paper. Additionally, there should be "Mean" instead of "Medium" in the heads of tables 5 - 7.

·         Tables 8., 9. and 10. should be moved to section 3.4. (for table 8) and 3.5. (for Tables 9 and 10, as well as Figure 2)

·         Figure 3. and Figure 4.  are placed in the main text before Figure 2.; additionally they are really difficult to analyse (even in 160% magnification), please make them more readable

·         we don't usually include links to results in the discussion, please remove them

some English grammar issues are present, so please read the paper carefully and check for them. I have marked some of them in the text, but because I am not a native speaker myself, there are probably others also

Round 2

Reviewer 1 Report

While the majority of questions have been satisfactorily addressed, there are certain aspects that warrant further consideration to enhance the robustness of this study. Specifically, the relatively lower number of enrolled COVID-19 patients and the omission of an analysis of SARS-CoV-2 variants with varying levels of virulence could be perceived as potential limitations. It is essential to address these factors comprehensively and elaborate on their potential impacts on the dynamics of leukocyte subpopulation changes during the progression of COVID-19.

Furthermore, I noticed that Figure 3 is conspicuously absent from the current revised manuscript (lines 313-314). I kindly request its reinstatement to augment the overall comprehensiveness of the presentation.  

Author Response

Please see the attach

Answers REVIEWER #1:

Thank you very much for all comments and suggestions.

We agree with the comment that the limitation of our study is the number of patients, especially at later points in time. We mention this in the discussion, lines 386 – 387.  The lack of information about SARS-CoV-2 variants is also a significant deficiency in this research. We tried to collect material from patients in the shortest possible time to create a fairly uniform group. Our diagnostics procedure indicate only the presence of the SARS-Cov-2 virus, and not its individual variants. This is a valuable consideration and in the future we will try to deepen the diagnostics.

Information about Figure 3 has been placed in lines 313-314 and 322. Figure 3 is visible under line 323. The old figure has been deleted. A new Figure 3 has also been added below in the comment.

Any revisions to the manuscript were marked up using the “Track Changes” function MS Word, such that any changes can be easily viewed.

We hope that the our changes and our answer will have a positive impact on your opinion about our work. Thanks’ your all above suggestions. We believe that they strengthened our work. Thank you kindly for your consideration.

ment